# Fighting Lead Poisoning: Effective Conditions for Home-Based Education, Housing Remediation, and Relocation [note 1]

**DOI:** 10.3390/toxics13070552

**Published:** 2025-06-29

**Authors:** Hugues de Barberin-Barberini, Elisabeth Jouve, Jean-Christophe Dubus, Karine Hadji, Remi Laporte

**Affiliations:** 1Service d’Accueil des Urgences Pédiatriques, Hôpital Nord, AP-HM, 13015 Marseille, France; 2Permanence d’Accès aux Soins de Santé Mère-Enfant, Hôpital Nord, AP-HM, 13015 Marseille, France; 3Service d’Evaluation Medicale, AP-HM, 13005 Marseille, France; 4Département Santé Environnement, Agence Regionale de Santé Provence-Alpes-Côte d’Azur, 13003 Marseille, France; 5Faculté de Médecine, Equipe de Recherche EA 3279 “Santé Publique, Maladies Chroniques et Qualité de Vie”, Aix Marseille University, 13007 Marseille, France

**Keywords:** childhood lead poisoning, substandard housing, slum, educational intervention, housing remediation, housing relocation

## Abstract

Background—Against childhood lead poisoning, removing lead exposure is the main measure, but how to do it effectively has not been fully established. Our objective was to determine the impact of several interventions (education, housing remediation, and relocation) on children’s blood lead levels. Methods—A historical cohort of childhood lead poisoning was drawn in Marseille, France, from 2011 to 2018. A generalized mixed model was developed to study the kinetics of blood lead levels. Results—We included 151 children, with 56% living in legal substandard housing and others living in slums. Medical follow-up (median: 612 days) included 492 blood samples. In legal substandard housing, blood lead level decrease was significantly associated with every intervention. In slums, blood lead level decrease was significantly associated with housing relocation and education, although to a lesser extent. Conclusions—Every intervention contributed to reducing blood lead levels in substandard housing. Educational intervention is rapidly implemented. Housing remediation follows a long-lasting but effective legal procedure. Some families get housing relocation, depending on their financial resources or whether they are eligible for social housing. In slums, access to legal housing is the most effective against environmental exposure and education has a wider impact on health literacy.

## 1. Introduction

The burden of lead poisoning in 2002 in Europe amounted to at least 1,053,000 disability-adjusted life years per year, mostly secondary to housing-based exposures [1]. In France, 417 new cases were diagnosed in 2020 (but probably under-diagnosed during the COVID-19 pandemic [2]). Childhood lead poisoning is defined as a lead blood level greater than 50 µg/L [3]. Lead paint has been banned for professional use since 1949, but it was authorized for sale until 1993 and persists in the environment. Vulnerable populations remain particularly at risk when living in old, substandard housing [4] or slums [5].

The protocol for managing childhood lead poisoning recommends medical follow-up to detect complications and surrounding cases and to monitor the decrease of blood lead levels. Environmental investigation and intervention to control the sources of lead release [6] and educational intervention to adopt healthy habits [7] are also required. The legal procedure governing environmental intervention calls for housing remediation work to be carried out when there is a source of lead exposure in the housing. Such remediation work is undertaken at the owner’s charge and must be carried out by taking specific precautions to avoid over-exposing residents [8]. Some families change housing on their own, but the legal procedure does not provide for specific access to social housing. This procedure is lengthy to put in place.

Advice on changing to healthy habits can be given quickly to start reducing lead exposure as much as possible [7].

Several studies have highlighted the impact of home-based educational intervention against housing-related environmental hazards. Shani has shown their effectiveness in childhood asthma in the USA [9]. In France, De Blay has shown that they improve compliance in controlling adult exposure to mite allergens [10]. Montaudié-Dumas has shown that they improve compliance in controlling children’s exposure to mite allergens and molds [11] (unfortunately, the effectiveness was insufficient to reduce exposure to tobacco smoke).

A systematic review of the literature questioned the effectiveness of educational and environmental interventions to control at-home lead exposure. It was initiated in 2008 and updated several times until 2020 [12,13]. The authors found educational intervention not effective in reducing blood lead levels and insufficient evidence to clarify whether environmental intervention reduces blood lead levels.

Only Randomized Controlled Trials (RCTs) and quasi-RCTs were analyzed. This restriction guaranteed a high level of scientific evidence. Unfortunately, in conditions where the fight against lead poisoning is already the subject of public policy [14], ethical and legal limitations reduce the power of RCTs to show persuasive results. The studies analyzed covered different interventions dating back to 1980. Impact outcome methods also varied widely: blood lead level after intervention (continuously measured or exceeding different thresholds: 100 or 150 µg/L after 6 to 18 months) and dust lead contamination (in household floor, household window). It was not clear whether sources of lead exposure were of the same kind, especially when processing soil abatement (rather appropriate for industrial pollution in brownfield sites). Lead risks in North America and Australia (regions of the studies reviewed) may also be different from Europe, limiting the scope of these results.

To our knowledge, there is no study investigating the impact of home-based educational intervention against lead poisoning in Europe. The aim of this study was to compare the effectiveness of different measures for controlling lead exposure in patients with childhood lead poisoning in France by analyzing the decline in blood lead levels.

## 2. Materials and Methods

This study adopted a monocentric historical observational cohort methodology, conducted in a specialized center, the Consultation Enfant-Environnement in Marseille, France, between November 2011 and December 2016. Patient management followed the 2006 national recommendations, and no changes were made to standard care [15].

Patients included had to be under 18 years of age, have an initial blood lead level of over 50 µg/L, live in the Bouches-du-Rhône area, have had a follow-up blood lead level (at least one subsequent control), and the result of an environmental investigation.

Data collected included age, sex, type of housing (legal or slum), pica–geophagy behavior, parental at-risk occupation, surrounding cases, associated substandard housing criteria (according to the criteria of the January 2002 French decree [16]: lack of solidity, too small surface area, lack of ventilation or natural lighting, sealing problems, other hazards: electrical, mechanical, pests and mold…), and sources of lead exposure.

An environmental investigation was conducted for each childhood lead poisoning case [6]. The type of housing was classified as legal for children living legally in a housing (renting or owning). Illegal substandard housing was classified as slum when it met UN-Habitat’s 2003 criteria [17] (including lack of access to water or sanitation, illegal occupation and risk of eviction, poor housing quality, and overcrowding). There was no sub-classification according to the various disorders. This would have greatly reduced the power and would have been a source of classification bias. These families shared a common insecurity and instability in their living conditions (often changing location, often risking having to do so at any time, and not always revealing each living place they use to healthcare providers).

Parental occupation was considered at risk, according to a published list [7].

Exposure data were collected from environmental survey results (paint, elemental lead, smoke and dust, or water risk) [6].

Three types of interventions were analyzed:

Home-based educational intervention aimed to promote the adoption of habits to be applied on a daily basis, was provided by a trained professional delivering advice taught by one of the French diplomas [18]) and delivered according to the recommendations of the French Ministry of Health [7]: potential sources of lead were identified in the children’s environment; prevention advice was adapted to the sources (including old paint, tap water, industrial hazards, and other sources) for children and adults (especially pregnant women); dietary advice were given; and actions and precautions to be taken by the family directly on the sources of exposure were detailed.

Housing remediation work aimed at eliminating all sources of lead in the environment (by evacuating) or making them out of reach (by sealing and covering).

Housing relocation was considered with regard to the risk of lead exposure, regardless of whether this was by the family’s own means or with social assistance. We considered that patients arriving in France for the first time and whose environmental investigation did not identify any source of lead exposure had a housing relocation on the date of arrival in France. Thus, housing relocation included children and their families moving from one legal housing to another, from abroad to legal housing, or from a slum to legal housing. Changes in slum sites were not taken into account (cf. supra).

Biological and clinical follow-ups were ideally carried out every three months until a blood lead level of less than 50 µg/L was achieved. To ensure continuous monitoring of siblings, even follow-up blood lead levels below 50 µg/L were included in the analysis for families with several children.

Patients who had not been seen for three months were regularly contacted by phone. Patients were excluded if the date of intervention was unknown.

The primary outcome was the blood lead levels according to the type of intervention in a multivariate model, taking into account time interaction and other confounding factors.

Patients were informed of the use of their personal and medical data for research purposes and their right to refuse.

Patients’ characteristics were compared between types of housing (legal housing vs. slum) using *t*-test for continuous variables and by chi2 or Fisher’s exact test for categorical variables with SPSS 20 software, IBM SPSS Statistics (Chicago, IL, USA). Lead kinetics were analyzed by type of housing using multivariate modeling using SAS Institute software 9.3, SAS (Cary, NC, USA). Blood lead levels were log-transformed. The model selected had a random intercept and slope for the time effect (repeated blood lead levels) and included fixed effects: age at inclusion, associated housing substandard criteria, type of intervention, time, time x intervention interaction. Model coefficients were calculated with standard errors. The means estimated by the model were calculated with standard errors and compared with the mean blood lead level before any intervention, with *p*-values given by Tukey’s adjustment.

## 3. Results

One hundred and sixty-five patients were included. Among them, 14 were excluded from the analysis in accordance with the criteria (unknown date of intervention). In all, 492 blood lead levels from 151 lead-poisoned patients were studied (Figure 1).

### 3.1. Initial Situation

The initial situation of lead-poisoned children is described according to their type of housing (Table 1). The sex ratio (1.60) and mean age 5.3 years (SD: 1.9) were comparable between groups. Children aged under 7 had significantly higher initial blood lead levels of 155.8 μg/L (SD: 35.0) than those aged 7 and over at 105.8 μg/L (SD: 17.5) (*p* < 0.005).

Children living in legal housing were mainly exposed to decayed paint and showed pica-like behavior. Very few cases of intoxication due to lead pipes (water contamination) were identified (only two patients from the same family). Children with pica and geophagy behavior were exclusively in legal housing.

Children living in slums had significantly higher blood lead levels (+27.5 μg/L; 95% confidence interval [5.7; 49.2]). The main sources of lead exposure were smoke and dust associated with parents’ occupations (burning and recycling generally in or close to the housing). Cases were grouped together, and living conditions were unhealthy (several families sharing the slum).

### 3.2. Follow-Up

The mean patient follow-up was 612 days (SD: 450). Patients were reviewed, on average, every 184 days (SD: 152). At the time of analysis, 84 (55.6%) of the children still had blood lead levels below 50 μg/L.

The median time for intervention (after the first blood lead above 50 µg/L) was 27 days (Interquartile = [0; 83]); 368 days (Interquartile = [267; 434] and 275 days (Interquartile = [219; 472]) for education, housing remediation, and housing relocation, respectively. The mean decrease in blood lead level in the interval from 3 to 5 months after intervention was measured for 45 children. This decrease was −39.7 µg/L (sd: 57.7), −109.0 µg/L (sd: 74.2), and −54.0 µg/L (sd: 28.6) after education intervention, housing remediation, and housing relocation, respectively (*p* = 0.06). In order to take into account all follow-up data on children’s blood lead levels, two different multivariate models were constructed independently of each other, given the significant differences between children in legal housing and in slums (Table 2). In both models, the youngest children had higher blood lead levels.

In legal housing conditions, associated housing substandard criteria were associated with higher blood lead levels. Each intervention (home-based education, housing remediation, housing relocation) led to a significant reduction in blood lead levels over time.

In slums, housing relocation was effective in significantly reducing blood lead levels. The home-based educational intervention was also significantly effective but with a smaller decrease enhanced. There was no remediation work in this group; procedural remediation was restricted to legal housing.

## 4. Discussion

These results have been presented in a conference of the The European Public Health Association [19]. Results showed a significant decrease in lead blood levels in legal housing after every intervention: home-based education, housing rehabilitation, and relocation. In slums, housing relocation was the most effective intervention in decreasing lead blood levels, followed by education. The main sources of lead exposure were different between legal housing (decayed paint) and slums (smoke and dust).

This study provides important real-world evidence in fighting childhood lead poisoning. Home-based educational intervention in legal housing aimed to control dust emissions from primary domestic sources of lead exposure. No sources were identified in the neighborhood. This educational intervention showed a significant impact, as in other studies about home-based educational intervention against housing-related environmental hazards (to reduce exposure to mite allergen, humidity, mold, volatile organic compounds, and outdoor triggers for asthma) [9,10,11].

The educational intervention used a single at-home individual interview. In other studies, educational intervention had often repeated sessions [20], as long as some used only multimedia [21].

Measuring the effectiveness of educational intervention requires taking into account the health literacy level, as did Shen [21] and other community-based interventions. This may explain the limited effectiveness of our educational intervention in slums. These families are facing multiple social and environmental vulnerabilities. They make priority choices more dictated by survival needs but also by their difficulties in accessing basic public services, including the healthcare system [22,23].

Environmental intervention in legal housing was considered to be aimed at changing the environment so that controlling exposure to lead sources did not depend on changing the family’s daily lifestyle habits (considered as an educational topic). As shown by Dixon, window replacement was effective in controlling lead paint hazards [24].

Our study showed another scale of the «cocktail effect» [25] in environmental hazards. Here, in substandard housing, there was the potentiation of lead exposure by other environmental multiple hazards (as defined by the criteria of the January 2002 French decree [17]). The impact of these other environmental multiple hazards failed to be depicted in slums.

Our results are different from the literature review by Nussbaumer-Streit [13]. However, among the reviewed studies, some interventions included primary prevention [26,27] (which often requires a larger sample size). High-Efficiency Particulate Air (HEPA) vacuuming [28] and assistance in cleaning [29] were classified as environmental interventions. We rather consider them as incentives in educational intervention (for quicker change to adopt healthy habits) [30]. Furthermore, another cleaning supply distribution was classified as an educational intervention [31].

In this review, targeted sources of lead exposure were wide-ranging, including some restricted to the housing and others in the neighborhood. Thus, soil abatement in Farrell’s study was not expected to be fully effective when the main source of lead exposure was decayed paint, and polluted soil was only a secondary co-factor of dust emission [32].

This literature review highlights how RCTs have limited power in studying interventions against lead poisoning. Indeed, RCTs are mostly designed to document the efficacy of a new product or action, with very restricted and controlled dissemination, and by looking for relatively short-term health effects (which lead is definitely not, when the aim is to eliminate exposure). Here, intervention is aimed at removing exposure to a well-known widespread toxicant. This may have limited the ways to design comparisons of challenger and control intervention with sufficient power while maintaining “the clinic equipoise” [33]. Each methodology either used sequential, time-limited intervention or compared intervention with control groups that were rarely the absence of any intervention at all.

Educational intervention is the quickest to organize. Several studies look at the repetition of educational intervention over time [20]. Indeed, although not documented, long-term loss of compliance is expected, particularly when no environmental intervention succeeds in controlling the source of lead exposure.

Access to social housing offers an alternative at least as effective as the current lead poisoning secondary prevention involving remediation work. Homes built before 1949 are required to have a Lead Exposure Risk Report carried out when they are put up for sale or rent, in accordance with French regulations [34]. Unfortunately, substandard and unhealthy housing is still poorly taken into account when allocating social housing in cases of childhood lead poisoning [35], given the shortage of social housing and despite the recognized enforceable right to housing, which imposes on the State an obligation of result in terms of access to decent housing [36].

Furthermore, housing relocation removes the child from sources of lead exposure in the environment. Unfortunately, the risk of lead poisoning remains when a new family moves in, if no rehabilitation work is carried out on the premises. The first way to avoid this risk in France is the requirement to show the tenant a Lead Exposure Risk Report [6], but this does not take full account of the risk of lead exposure (excluding buildings after 1949 and water hazards). A new approach would be to make Prior Authorization to Rent out a Property conditional on the elimination of any risk of lead exposure [37]. Unfortunately, this authorization is only required in some areas and, so far, does not cover this risk (even if it has already been identified for the property).

In slums, burning and recycling are major and hardly controlled sources of lead exposure. For many families, this is the only source of income. Families choose to prioritize economic survival over environmental and health protection. Children living in these contexts are thus exposed to lead daily, but also to multiple pollutants that are not sought after in dust and fumes, with massive air pollution [38] and long-term soil pollution [39]. Pointing out the risk of lead poisoning in this illegal housing provides a paradoxical argument for enforcement to evacuate these areas while many families are offered no alternative. However, among interventions for youth experiencing homelessness, family housing is one of the most promising interventions against current and future environmental risks [40].

Sources of lead exposure should be the primary stratification argument in a review of studies rather than the type of intervention. As highlighted by many, including Boreland, it is important to assess potential sources of lead exposure and pathways by which children are exposed to lead when studying a risk reduction intervention [41].

Improving health literacy is a priority to mediate the relationship between highly deprived socioeconomic status and health inequalities, among which lead poisoning is only one of many risks [42,43,44,45]. In slums, priority educational issues need to rather focus on health mediation intervention to empower the families with health literacy [46] and improve their social and economic capital.

The methodology of our study had inherent limitations in establishing causal inference, but it provided an argument in supporting the temporal association criterion between several interventions and lead blood level lowering. Temporal association is one of the nine criteria required to establish causality in epidemiology, according to Hill [47]. This study is a retrospective observational cohort, as the standard care protocol was followed as recommended (no randomization), although prolonged delays are to be noted compared with recommendations [6]. Delays depended on a number of uncontrolled confounding factors (but to know the situations, much more related to many administrative delays than to family-specific factors). Socioenvironmental and cultural confounding factors cannot be excluded, but they should be frequent and homogeneous enough to cause bias (i.e., confounding factors need a systematic double association with the benefiting of one intervention and the decrease of blood lead level—independently from the effect of the intervention itself).

Nevertheless, this cohort, following a systematic management protocol, was better able to integrate data from children with reduced compliance to follow-up, taking time into account, compared to an RCT that would have rapidly excluded them [48]. Several reviews of RCTs failed to furnish consistent strategies and guidelines against childhood lead poisoning [13,48]. Clinicians were left alone in front of this public health issue, and many children were in front of uncontrolled lead exposure. Those conclusions call into question the improvement in the medical service thus rendered. Moreover, Jacobs has pointed out that studies showing the efficacy of some interventions may have been overlooked [49].

In public health, evidence from non-RCTs may provide a more adequate or best available measure of impact [50]. Non-RCT evidence is often considered as lower quality. However, the Grading of Recommendations Assessment, Development, and Evaluation approach to grading the quality of evidence and strength of recommendations considers that Hill’s criteria enable to establish causation. Furthermore, several enhancements to these criteria have been published [51].

Now, massive health data epidemiology will help to overcome the challenge of comparing the effectiveness of different interventions in real life. The impact of each intervention type can be assessed by adjusting for selection biases, for example, via propensity score matching [52]. This approach requires consideration of the inherent limitations of observational studies and validating the balance of covariates, but massive health data epidemiology offers a promising way forward compared with the disappointing results of past RCTs.

## 5. Conclusions

As with any social health inequity, childhood lead poisoning requires a complex strategy involving and targeting the living environment and social situation of the family.

In substandard housing, every intervention (home-based education, housing remediation, and relocation) was effective in decreasing blood lead levels. Educational intervention is the quickest way to control exposure. Housing remediation work is part of a legal procedure, which is long but effective. Some families manage to relocate their housing, depending on their financial resources or whether they are eligible for social housing for other reasons, but this is also very time-consuming.

In slums, educational intervention can primarily improve families’ health literacy and their social and economic capital. Wherever possible, access to legal housing should be targeted for its multiple positive impacts on the health and well-being of the whole family, including the fight against lead poisoning.

## Figures and Tables

**Figure 1 toxics-13-00552-f001:**
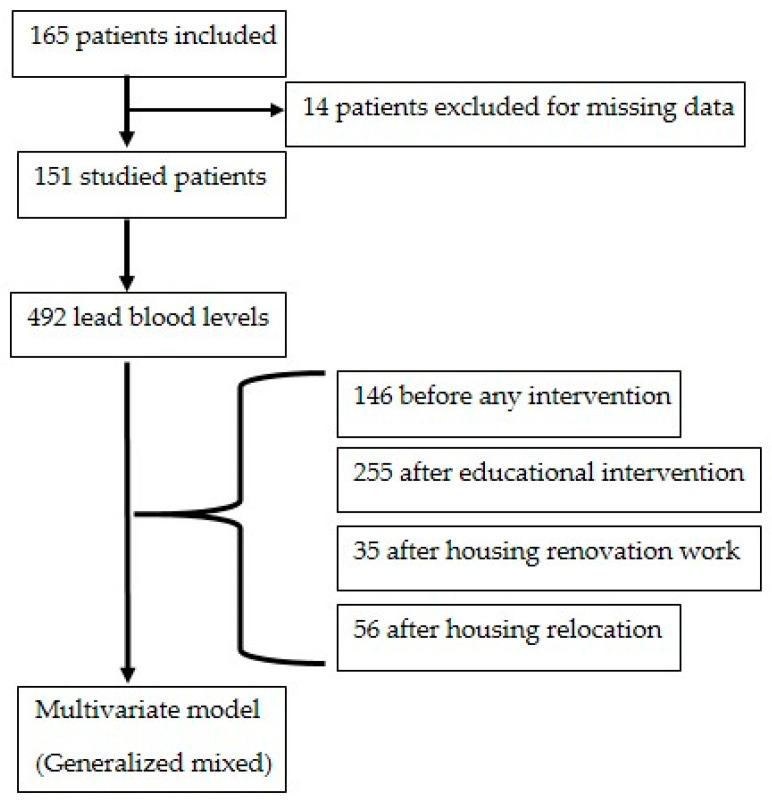
Flow chart.

**Table 1 toxics-13-00552-t001:** Patient description.

		Legal Housing	Slum	*p*
N (%)		85 (56.3)	66 (43.7)	
Boys		52 (61.2)	41 (62.1)	0.13
Age at diagnosis (years) *		4.9 (1.7)	5.9 (2.0)	0.14
Sources of lead exposure				
	Decayed paint	63 (74.1)	7 (10.6)	<0.0005
	Smoke and dust (burning, recycling)	21 (24.7)	63 (95.5)	<0.0005
	Polluted water (leaded pipes)	2 (3.0)	0 (0.0)	0.51
Risk co-factors				
	Pica–geophagy behavior	10 (11.8)	0 (0.0)	0.005
	Surrounding cases	77 (90.6)	63 (95.5)	0.35
	Associated housing substandard criteria	44 (51.8)	57 (86.4)	<0.0005
	Parental at-risk occupation	15 (17.6)	54 (81.8)	<0.0005
Blood lead level at diagnosis (μg/L) *		110.7 (34.5)	138.4 (32.2)	0.01

* mean (standard deviation).

**Table 2 toxics-13-00552-t002:** Effects of intervention against sources of lead exposure (linear mixed model).

Housing Type		Legal		Slum	
		Coefficient (Standard Error)	*p*	Coefficient (Standard Error)	*p*
Intercept		71.4 (12.9)	<0.0001	105.8 (11.4)	<0.0001
Inclusion age < 7 years		19.2 (12.0)	0.11	47.7 (13.7)	0.001
Associated housing substandard criteria		36.6 (11.3)	<0.005	-	
Time (months)		4.3 (1.54)	<0.01	1.0 (0.94)	0.29
Interaction time x intervention, after:					
	Home-based educational intervention	−5.4 (1.52)	<0.001	−2.1 (0.82)	0.01
	Housing remediation	−6.0 (1.57)	<0.001	-	
	Housing relocation	−5.7 (1.56)	<0.001	−4.5 (1.0)	<0.0001

## Data Availability

The data presented in this study are available on request from the corresponding author.

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
