# Peer review of "Fighting Lead Poisoning: Effective Conditions for Home-Based Education, Housing Remediation, and Relocation†"

_toxics, 2025, doi:10.3390/toxics13070552_

Round 1

Reviewer 1 Report

Comments and Suggestions for Authors

This is a valuable and clinically relevant study addressing a critical public health issue – childhood lead poisoning in vulnerable populations. It provides important real-world evidence on the effectiveness of different interventions (education, remediation, relocation) in distinct housing contexts (legal substandard vs. slums) in Marseille, France. The manuscript is generally well-written and structured. However, several limitations should be addressed to strengthen its impact and generalizability.
1. Causality and Confounding: The observational design inherently limits causal inference. Interventions weren't randomly assigned, and timing/selection likely depended on factors also influencing blood lead level decline (e.g., family motivation, severity, social services engagement, unmeasured environmental changes).
2. Quantifying the "Educational Intervention": The description of the educational intervention is somewhat generic ("advices taught by one of the French diplomas," "promote adoption of habits"). Its effectiveness, especially the differential effect between housing types, needs deeper exploration.
3. Limited Detail on "Slum" Context and Relocation: While relocation was most effective in slums, details on what relocation entailed (type of housing, location relative to previous exposure sources, duration of stay) and how families achieved it (social housing, self-funded, NGO-assisted?) are sparse. This limits understanding of the key factors for success.
4. line51:  “lenghty” should be “lengthy”.
HEPA abbreviation is put in the end (line 302). It is better to define it immediately

Author Response

We are very grateful for this review and comments. Your comments were very fair and interesting, and enabled us to improve considerably our article. Please find enclosed our revised manuscript as requested. We have made changes and provided additional information on each issue raised.

Comment 1 : 1. Causality and Confounding: The observational design inherently limits causal inference. Interventions weren't randomly assigned, and timing/selection likely depended on factors also influencing blood lead level decline (e.g., family motivation, severity, social services engagement, unmeasured environmental changes).
Response 1: Randomization is basically a clinical research method, and not the only one that can establish recognized causality. Epidemiological inference is well described according to causality criteria published by Hill in 1965 (and updated since). We have included these elements in the discussion.
Revised manuscript page 8 line 304 : " The methodology of our study has inherent limitations in establishing causal in-ference. But it provided an argument in supporting the temporal association criterion between several interventions and lead blood level lowering. Temporal association is one of the nine criteria required to establish causality in epidemiology according to Hill [47]. This study is a retrospective observational cohort, as the standard care pro-tocol was followed as in recommended standard care (non-randomized), although prolonged delays are to be noted compared with recommendations[6]. Delays de-pended on a number of uncontrolled confounding factors (but to know the situations, much more related to many administrative delays than to family-specific factors). So-cio-environmental and cultural confounding factors cannot be excluded, but they should have be frequent and homogeneous enough to cause bias (i.e. confounding fac-tor need a systematic double association with benefiting of one intervention and the decrease of blood lead level – independently from the effect of the intervention itself). "

Comment : 2. Quantifying the "Educational Intervention": The description of the educational intervention is somewhat generic ("advices taught by one of the French diplomas," "promote adoption of habits"). Its effectiveness, especially the differential effect between housing types, needs deeper exploration.
Response 2 : We included the description in the text according to your comment.
Revised manuscript page 3 line 113 : " Home-based educational intervention aimed to promote the adoption of habits to be applied on a daily basis, by a trained professional delivering advices taught by one of the French diplomas[19]),and was delivered according to the recommendations of the French Ministry of Health[18]: potential sources of lead were identified in the chil-dren’s environment; prevention advice was adapted to the sources (including old paint, tap water, industrial hazards and other sources) for children and adults (espe-cially pregnant women); dietary advice were given; and actions and precautions to be taken by the family directly on the sources of exposure were detailed."

Comment : 3. Limited Detail on "Slum" Context and Relocation: While relocation was most effective in slums, details on what relocation entailed (type of housing, location relative to previous exposure sources, duration of stay) and how families achieved it (social housing, self-funded, NGO-assisted?) are sparse. This limits understanding of the key factors for success.
Response 3: We included the description of the different relocation process in the text according to your comments
Revised manuscript page 3 line 123 : " Housing relocation was considered with regard to the risk of lead exposure, re-gardless of whether this was by family’s own means or with social assistance. We con-sidered that patients arriving in France for the first time and whose environmental in-vestigation did not identify any source of lead exposure had a housing relocation on the date of arrival in France. Thus, housing relocation included children and their fam-ily moving from a legal housing to another, from abroad to a legal housing, or from a slum to a legal housing. Changes of slum site were not taken into account (cf. supra)."

Comment 4 : line51: “lenghty” should be “lengthy”.
Response 4 : We have made the correction according to your comment.
 Revised manuscript page 2 line 51 ;

Comment 5 : HEPA abbreviation is put in the end (line 302). It is better to define it immediately
Response 5 : We have made the correction according to your comment.
Revised manuscript page 7 line 240 : " High-Efficiency Particulate Air (HEPA) vacuuming "

We hope that these answers meet your expectations and that the manuscript now suits you, and we remain at your disposal should you require any further clarification.
Best regards

Reviewer 2 Report

Comments and Suggestions for Authors

Review of toxics-3694330-peer-review-v1

This is an excellent paper. Thank you for the opportunity to review it. The English needs improvement throughout.

Line 20. The use of the word “slums” should be clarified: Are these illegal substandard homes or something else?

Line 22. “interventions” here and throughout should be singular, not plural. Also, it would help to report the mean blood lead reduction, not only what the reductions were associated with.

Line 36. The case definition of lead poisoning in France is needed here. Also, the year lead paint was banned in 1949 pertains to France, but is sadly still available in other countries.

Line 51. Although it may be true that education can be delivered more quickly than remediation, education does depend on compliance, and there may be some sources of exposure that many not be apparent to occupants (invisible small dust particles for example).

Line 69. This reference is also germane to the discussion here: Jacobs DE. Lead screening update from the US Preventive Services Task Force. J Pediatr. 2019 Sep;212:243. doi: 10.1016/j.jpeds.2019.07.011. PMID: 31439165. This article said in part: “Commentary Updating best practices using available high quality

evidence is an important and worthy undertaking.

Sadly, this update fails to inform best clinical practice and if followed,

not only places clinicians at significant liability, but

more importantly, needlessly abandons children to lead exposures

that are entirely preventable. This is precisely why the US

Preventive Services Task Force recommendation is at odds

with nearly every other consensus statement on the subject,

including the Centers for Disease Control and Prevention,1

American Academy of Pediatrics,2 and many others.3 Simply

put, this recommendation should not be followed. Furthermore,

clinician failure to screen high risk children has been

the subject of repeated investigations by the Government

Accountability Office and Congressional hearings. Almost

none of the key studies in the field of lead screening, hazard

identification, and control were cited in the Task Force’s recent

assessment. These are reviewed in detail elsewhere.4 For

example, a randomized controlled trial found that among

non-Hispanic black children, blood lead concentrations were

31% lower (95% CI, _50% to _5%; P = .02) in the intervention

group than the control group.5 The Task Force failed to

include this and dozens of other studies showing significant reductions

in blood and dust lead levels following screening and

hazard control, including a large-scale study of 14 jurisdictions

and nearly 3000 housing units.6 Historically, clinicians have

appropriately ignored the Task Force’s previous recommendations

in this area because their review is viewed as flawed and

incomplete. This is evidenced by the fact that nearly 3 million

children had blood lead levels reported to the CDC in 2016.7 With over half a million children with blood lead levels above

the CDC reference level and with over 37million housing units

harboring lead paint, lead exposure remains a large pressing

problem that requires action, not just calls for more evidence.

The practice of ignoring the Task Force’s recommendations

will likely continue with this latest update. One can only

hope that future updates will include those who are more

knowledgeable in the field and the many studies that were

not included in this most recent review. Proven best practice

would involve taking action to prevent exposures, screening

of high risk children, and then referring families who have children

with elevated blood lead levels to risk assessors and others

who are trained and licensed to identify, quantify, and remediate

exposure. The evidence supporting this best practice is

best detailed in the CDC Advisory Committee Statement.3 Clinicians

should not suggest that reliable information on how to

identify and control lead hazards is “unavailable” because this

does not reflect the current science. Lead content in millions of

homes have been successfully abated, which in part explains

why blood lead levels have declined. Clinicians should be

part of this successful effort and not ignore the evidence.

Line 117. Perhaps “confusion” should be “confounding” in this sentence?

Line 169. Although relocation was identified as a significant predictor, the authors might also point out that if the previous house was not remediated (as is often the case during relocation), then a subsequent child could be exposed.

Line 262. This sentence says the study was “non-intervention” but there was in fact some interventions, such as education and remediation and relocation, so it is not clear why the authors considered this non-interventional. This could be clarified.

End

Comments on the Quality of English Language

The English can be improved in this paper.

Author Response

We are very grateful for this review and comments. Your comments were very fair and interesting, and enabled us to improve considerably our article. Please find enclosed our revised manuscript as requested. We have made changes and provided additional information on each issue raised.

Comment 1 : The English needs improvement throughout.
Response 1 : We apologize for our poor English. As the two reviewers disagreed on this point, we assumed “minor corrections” were of concern. The editor proposes to correct them in the editing process, and we hope that this will meet with your satisfaction.

Comment 2 : Line 20. The use of the word “slums” should be clarified: Are these illegal substandard homes or something else?
Response 2 : We included the definition in the text according to your comment.
Revised manuscript page 3 line 101 : " Illegal substandard housing was classified as slum when it met UN-101 Habitat's 2003 criteria[17] (including lack of access to water or sanitation, illegal occupa-102 tion and risk of eviction, poor housing quality, and overcrowding). There was no sub-103 classification according to the various disorders. This would have greatly reduced the 104 power and would have been a source of classification bias. These families shared a com-105 mon insecurity and instability in their living conditions (often changing location, often 106 risking having to do so at any time, and not always revealing each living place they use 107 to healthcare providers)."

Comment 3.1 : Line 22. “interventions” here and throughout should be singular, not plural. Also, it would help to report the mean blood lead reduction, not only what the reductions were associated with.
Response 3.1 : We reworded the text according to your comment.
Response 3.2 : We have included this result in the text in line with your comment, although consistent results can only include a subgroup of patients (those whose blood lead monitoring was performed at the same time, here 3 months), as shown below.
Revised manuscript page 5 line 187 : " The mean decrease in blood lead level in the interval from 3 to 5 months after an intervention was measured for 45 children. This decrease was -39.7 µg/l (sd: 57.7); -109.0 µg/l (sd: 74.2) and -54.0 µg/l (sd: 28.6), after education intervention, housing remediation and housing relocation respectively (p = 0.06). In order to take into account all follow-up data on children's blood lead levels, two different multivariate models were constructed independently of each other, given the significant differences between children in legal housing and in slums (Table 2). "

Comment 4 : Line 36. The case definition of lead poisoning in France is needed here. Also, the year lead paint was banned in 1949 pertains to France, but is sadly still available in other countries.
Response : We included the definition in the text according to your comment.
Revised manuscript page 1 line 37 : " Childhood lead poisoning is defined as a lead blood level greater than 50 µg/l[3]. " 

Comment 5 : Line 51. Although it may be true that education can be delivered more quickly than remediation, education does depend on compliance, and there may be some sources of exposure that many not be apparent to occupants (invisible small dust particles for example).
Response : We are aware of the partial, potentially temporary impact of education. This is mentioned in the discussion, and however the evaluation criterion is an impact indicator that takes these factors into account.
Revised manuscript page 6 line 221 : " Measuring the effectiveness of educational intervention requires taking into account 221 the health literacy level, as did Shen[21] and other community-based interventions. This 222 may explain the limited effectiveness of our educational intervention in slum. These fam-223 ilies are facing multiple social and environmental vulnerabilities. They make priority 224 choices more dictated by survival needs, but also by their difficulties in accessing basic 225 public services, including healthcare system[22,23]. "
Revised manuscript page 7 line 259 : " Educational intervention is the quickest to organize. Several studies look at the rep-259 etition of educational intervention over time[20]. Indeed although not documented, long-260 term loss of compliance is expected, particularly when no environmental intervention suc-261 ceeds in controlling the source of lead exposure. »

Comment 6 : Line 69. This reference is also germane to the discussion here: Jacobs DE. J Pediatr. 2019
Response : We fully agree with this comment. We have included this reference and highlighted the impact on clinical practice and strategic research perspectives.
Revised manuscript page 8 line 319 : " Several review of RCTs failed to furnish consistent strategy and guidelines against childhood lead poisoning[13,48]. Clinicians were left alone in front of this public health issue and many children in front of uncontrolled lead exposure. These conclusions call into question the improvement in the medical service thus rendered. Moreover, Jacobs has pointed out that studies showing the efficacy of some interventions may have been overlooked[49]. 
In public health, evidence from non-RCT may provide more adequate or best available measure of impact [50]. Non-RCT evidence is often considered as lower quality. However, the Grading of Recommendations Assessment, Development and Evaluation approach to grading the quality of evidence and strength of recommendations considers Hill’s criteria for causation. Furthermore, several enhancements to these criteria have been published [51]. "

Comment 7 : Line 117. Perhaps “confusion” should be “confounding” in this sentence?
Response : We reworded the text according to your comment.

Comment 8 : Line 169. Although relocation was identified as a significant predictor, the authors might also point out that if the previous house was not remediated (as is often the case during relocation), then a subsequent child could be exposed.
Response : Indeed, this impact is highly significant for the public health and had been omitted in discussion.
Revised manuscript page 7 line 272 : " Furthermore, housing relocation removes the child from sources of lead exposure in the environment. Unfortunately, the risk of lead poisoning remains when a new family moves in, if no rehabilitation work is carried out on the premises. The first way to avoid this risk in France is the requirement to show the tenant a Lead Exposure Risk Report [6]. But this does not take full account of the risk of lead exposure (excluding building after 1949, and water hazard). A new approach would be to make Prior Authorization to Rent out a Property conditional on the elimination of any risk of lead exposure[37]. Unfortunately, this authorization is only required in some areas, and so far does not cover this risk (even if it has already been identified for the property). "

Comment 9 : Line 262. This sentence says the study was “non-intervention” but there was in fact some interventions, such as education and remediation and relocation, so it is not clear why the authors considered this non-interventional. This could be clarified.
Response : Indeed « non-intervention » was confusing. We have reworded to distinguish between the methodology of an « observational study » and the different types of « intervention » against lead exposure.

We hope that our  answers will meet your expectations and that the manuscript now suits you, and we remain at your disposal should you require any further clarification.
Best regards